# Transcriptome Analysis of Rice Embryo and Endosperm during Seed Germination

**DOI:** 10.3390/ijms24108710

**Published:** 2023-05-13

**Authors:** Heng Zhang, Guang Chen, Heng Xu, Sasa Jing, Yingying Jiang, Ziwen Liu, Hua Zhang, Fulin Wang, Xiangyang Hu, Ying Zhu

**Affiliations:** 1State Key Laboratory for Managing Biotic and Chemical Threats to the Quality and Safety of Agro-Products, Key Laboratory of Traceability for Agricultural Genetically Modified Organisms, Ministry of Agriculture and Rural Affairs, Institute of Virology and Biotechnology, Zhejiang Academy of Agricultural Sciences, Hangzhou 310021, China; 2Shanghai Key Laboratory of Bio-Energy Crops, Research Center for Natural Products, Plant Science Center, School of Life Sciences, Shanghai University, Shanghai 200444, China

**Keywords:** rice, seed germination, endosperm and embryo, unfolded protein response (UPR)

## Abstract

Seed germination is a complex, multistage developmental process that is an important step in plant development. In this study, RNA-Seq was conducted in the embryo and endosperm of unshelled germinating rice seeds. A total of 14,391 differentially expressed genes (DEGs) were identified between the dry seeds and the germinating seeds. Of these DEGs, 7109 were identified in both the embryo and endosperm, 3953 were embryo specific, and 3329 were endosperm specific. The embryo-specific DEGs were enriched in the plant-hormone signal-transduction pathway, while the endosperm-specific DEGs were enriched in phenylalanine, tyrosine, and tryptophan biosynthesis. We categorized these DEGs into early-, intermediate-, and late-stage genes, as well as consistently responsive genes, which can be enriched in various pathways related to seed germination. Transcription-factor (TF) analysis showed that 643 TFs from 48 families were differentially expressed during seed germination. Moreover, 12 unfolded protein response (UPR) pathway genes were induced by seed germination, and the knockout of *OsBiP2* resulted in reduced germination rates compared to the wild type. This study enhances our understanding of gene responses in the embryo and endosperm during seed germination and provides insight into the effects of UPR on seed germination in rice.

## 1. Introduction

Seed germination begins with the uptake of water by dry quiescent seeds and ends with radicle protrusion, which is a complex developmental process that is a prerequisite for plant development and crop yield [1]. The process of seed germination is accompanied by complex physiological changes and energy metabolism, such as the decomposition of stored matter, energy generation, signal transduction, and transcription reprograming [2]. Rice is one of the most important food crops in the world and is the principal food of nearly half of the world’s population [3]. The large-scale planting of direct-seeded rice necessitates more attention towards the cultivation of rapid germination and vigorous seedling growth of rice [4]. Therefore, further investigation of the genetic and molecular mechanism of the rice-seed germination process is an important objective of direct-seeded rice breeding. In recent years, multi omics approaches have been used to elucidate the molecular mechanisms of rice-seed germination [2,5]. 

Water uptake by a mature dry seed is triphasic, (i) rapid initial uptake (phase I); a plateau phase (phase II); and (iii) a further increase in water uptake after germination is completed (phase III) [6]. In phase I, the genes of the receptor kinase signaling pathway, cell-wall degradation and modification, abiotic stress responses, as well as antioxidant-related genes, and many others, are activated [7]. At the beginning of phase II (24 h imbibition), seed maturation- and desiccation-associated protein degradation occurs [6]; in the middle of phase II (33 h imbibition), the imbibition response proteins, involving energy metabolism, cell growth, cell defense, storage proteins, and others are activated [8]; and at the end of phase II (48 h imbibition), the storage proteins are degraded [6]. However, these studies were mainly based on the germination process of shelled seeds, and most did not distinguish the transcription changes between embryos and endosperms at different stages. Little is known about the changes in gene expression in the embryo and endosperm of unshelled rice seeds during germination. 

Seed germination is regulated by both internal factors such as soluble sugar and external factors such as water availability, temperature, and plant hormones play important roles in coordinating these signals [5,9]. Abscisic acid (ABA)and gibberellins (GAs) are two key phytohormones that antagonistically regulate seed germination, with ABA suppressing and GAs enhancing seed germination [10,11,12]. ABA inactivates the type 2C protein phosphatases (PP2Cs) by binding pyrabactin-resistant/PYR-like/regulatory components of the ABA receptor (PYR/PYL/RCAR), allowing for the autophosphorylation and activation of snf1-related kinase 2 (SnRK2), which in turn phosphorylates bZIP transcription-factors (TFs) of the ABI5/AREB/ABF family; these TFs bind to ABA-responsive elements (ABRE) in the promoter sequence of ABA-inducible genes to positively regulate seed dormancy [13,14,15,16,17,18]. In *Arabidopsis*, the loss-of-function mutant in *ABA INSENSITIVE3* (*ABI3*), a central TF in the ABA signaling network, resulted in decreased sensitivity to ABA during seed germination [19]. Moreover, *ABI4* can decrease seed germination by interacting with RbohD and VTC2 and promoting ROS accumulation under salinity stress in *Arabidopsis* [20]. *ABI5* also plays a central role in the inhibition of seed germination while the interaction between BES1, a central-component TF of brassinosteroid signaling, and ABI5 suppresses the binding of ABI5 to the downstream gene-promoter region, which releases its inhibition on seed germination [21]. GAs can promote the rupture of endosperm during seed imbibition, and its biosynthesis mutants cannot germinate without the application of GAs [22]. DELLA proteins are important regulators of GAs signaling; for example, RGL2, which can promote ABA biosynthesis and inhibit seed germination [23]. Other plant hormones mainly regulate seed germination via crosstalk with ABA or GAs [9]. The early germination phenotype of auxin response factor *arf10* and *arf16* mutants was caused by the downregulation of the ABA-signaling TF *ABI3* [9]. Soluble sugar is another important signaling molecule involved in seed germination. Rice seeds contain a large amount of starch in endosperm for germination; α-amylase was proven to be the first enzyme to be attached to starch granules [24]. During seed germination, α-amylase breaks down starch into soluble glucose, which provides material and energy for seed germination. It was reported that the expression of α-amylase is induced by GAs and a lack of sugar [25]. The TF GAMYB can bind to GA-responsive elements in the α-amylase promoter and activate its expression in rice [26]. At the same time, the activation of α-amylase by GAs is inhibited by ABA signaling through protein kinase PKABA1 [27]. 

Embryo–endosperm communication plays a critical role in seed development and germination. In dormant seeds, the plant hormones (i.e., ABA) from the endosperm maintain embryo dormancy [28]. Some studies have shown that GAs in the endosperm during seed germination originate from the embryo [29] but there is also evidence that GAs can be produced in the endosperm [30]. The cell-death progress and α-amylase production from the embryo-containing zone to more distant zones of the endosperm indicate that the embryo promotes endosperm cell death during seed germination [28,31]. In addition, the mechanical interaction between the embryo and the endosperm also plays an important role in seed germination [28]. For example, endosperm softening and cell expansion of the endosperm cause the endosperm to rupture, allowing for embryonic root growth during germination [32]. Although many studies have shown that the interaction between the endosperm and embryo is very important in the process of seed germination, the molecular mechanism of the interaction is not fully understood.

In eukaryotes, approximately one-third of secreted and membrane proteins need to enter the endoplasmic reticulum (ER) for modification after translation, and then enter the corresponding organelles to perform their biological functions [33]. When eukaryotes were stressed, unfolded proteins accumulated in the ER and altered the gene expression to alleviate the excessive accumulation of unfolded proteins in the ER, for example, binding protein (BiP), a chaperone protein involved in the folding of secretory proteins in the ER lumen [34]. This phenomenon is called ER stress response or unfolded protein response (UPR), and the process is conserved in eukaryotic cells [35]. The UPR is involved in a variety of biological processes in plants. For example, *OsbZIP74* and *OsNTL3* are constitutively expressed and upregulated by heat and ER stresses, and coregulate the heat responses in rice [36]. UPR has also been shown to play a role in regulating seed development. For instance, in rice, overexpression of *OsrAAT* (*Oryza sativa* recombinant alpha antitrypsin) in the rice endosperm activates UPR and causes chalkiness in the rice grain [37]. Knockout of *OsbZIP60* or overexpression of *OsbZIP50*, *OsBiP1*, *OsBiP2*, and *OsBiP3* causes various degrees of chalkiness [38]. Overexpression of *OsBiP1* can inhibit the accumulation of seed-storage proteins in endosperm cells [39,40]. Although the regulatory mechanism of ER stress regulators has been well elucidated in rice, it is unclear whether UPR is involved in rice-seed germination. 

In this study, RNA-Seq was conducted in the embryo and endosperm of unshelled seeds germination. A Kyoto Encyclopedia of Genes and Genomes (KEGG) analysis showed that the pathways of differentially expressed genes (DEGs) enrichment in embryo and endosperm were not completely consistent. Embryo-specific DEGs were enriched in the plant-hormone signal-transduction pathway while the endosperm-specific DEGs were enriched in phenylalanine tyrosine and tryptophan biosynthesis. We also found that the expression levels of genes associated with the UPR pathway also changed in the embryo and endosperm during seed germination, for example, *OsBiPs*. Knockout of *OsBiP2* resulted in slow seed germination in rice. Our results suggest that the embryo and endosperm undergo different development processes during seed germination and the UPR pathway is involved in the regulation of rice-seed germination.

## 2. Results

### 2.1. Transcriptome Sequencing and Differentially Expressed Gene Analysis

To determine the difference in gene responses in embryos and endosperms during seed germination in rice, the unshelled rice seeds were imbibed with double-distilled water and sampled at 0, 1, 3, 6, 9, 12, and 24 h after treatment. Based on the seed imbibition process, we divided the seed-germination process into three stages: the first stage is associated with rapid water uptake (early stage, 1–3 h), the second stage is associated with embryo expansion (intermediate stage, 6–9 h), and the third stage is the beginning of germination when the radicle begins to grow (late stage, 12–24 h) (Figure 1). The embryos and endosperms of all-the-above germinating seeds were collected for RNA sequencing. We performed principal-component analysis (PCA) using DEGs and found that PC1 explained 44.64% of the total variation and separated the samples clearly between the different germination stages (Figure 2A). PC2 explained 24.81% of the total variation, which separated the samples between endosperms and embryos (Figure 2A). In total, 14,391 DEGs were identified between dry seeds and imbibed seeds. Among them, 7109 were identified in both the embryos and endosperm, 3953 were embryo-specific DEGs, while 3329 DEGs were endosperm specific (Figure 2B,D). The embryo-specific DEGs were enriched in the plant-hormone signal-transduction pathway, and endosperm-specific DEGs were enriched in phenylalanine tyrosine and tryptophan biosynthesis (Figure 2C). The DEGs enriched in the biosynthesis of amino acids, amino sugar and nucleotide sugar metabolism, glycolysis/gluconeogenesis, fatty acid biosynthesis, and biotin metabolism pathways were found in both the embryo and the endosperm during seed germination (Figure 2C). 

### 2.2. KEGG Enrichment Analysis of DEGs at Different Germination Stages

We classified the DEGs on a temporal scale to explore the molecular regulatory models of gene expression during seed germination. We observed that the number of DEGs increased along with the seed’s germination (Figure 3). At the early stage of seed imbibition (0–3 h), only 394 DEGs were found, with 287 genes differentially expressed in the embryo and 107 genes in the endosperm, respectively, indicating the distinct modes in molecular regulation between the embryo and endosperm (Figure 3A,C). With the proceeding of seed germination, more genes were differentially expressed, with 926 DEGs in the endosperm and 270 DEGs in the embryo, and 86 DEGs were found in both tissues at 6 h, suggesting that the major molecular changes take place 6 h during germination (Figure 3A,D). When it comes to the late stage, there were 822 DEGs found in the endosperm and 594 DEGs in the embryo; only 6 DEGs were common in both (Figure 3A,E), which means that the two parts of seeds undergo different biological processes. 

A total of 3341 genes were found to be consistently differentially expressed during seed germination, containing 1269 embryo-specific DEGs and 928 endosperm-specific DEGs (Figure 3A,B). KEGG analysis was performed with these genes, which are potentially important players during seed germination. Cluster 1 corresponded to 608 transcripts, whose accumulation profiles presented a rising trend in both embryos and endosperm, and which were enriched in amino acid and nucleotide sugar metabolism, phenylpropanoid biosynthesis, and diterpenoid biosynthesis pathways (Figure 3A and Appendix A). A total of 712 DEGs were clustered into cluster 2, whose expression levels kept rising and were much higher in the endosperm (Figure 3A). These DEGs are enriched to starch- and sucrose-metabolism pathways (Appendix A). In cluster 3, 946 DEGs remained highly expressed in the embryo at all stages, which were enriched to amino acid and nucleotide sugar metabolism, phenylpropanoid biosynthesis, and plant-hormone signal transduction (Figure 3A and Appendix A). Cluster 4 contained 800 DEGs, with the profile showing rising at the early stage and then decreasing along with the time course (Figure 3A). These kinds of expression-module genes were mainly classified as plant–pathogen interaction, alpha–linolenic acid metabolism, glutathione metabolism, and the MAPK signaling pathway (Appendix A).

### 2.3. Analysis of Differentially Expressed Transcription Factors

TFs are key regulators in plant development and defense. In this study, we identified 643 TFs that were differentially expressed during seed germination, which were classified into 48 families (Figure 4). MYBs, C2H2s, and CAMTAs were the families that had the most DEGs in all TF families, followed by bHLHs, NACs, and WRKYs, implying their key roles in rice-seed germination (Figure 4A). The expression pattern of over half (50.32%) of TFs could not be clustered to a specific germination stage, while 32.35% of them were consistently differentially expressed during imbibition (Figure 4B). As for the rest, 4.67% of TFs were differentially expressed at the early stage, 5.13% at the intermediate stage, and 7.62% at the late stage of seed imbibition, respectively (Figure 4B).

### 2.4. Hormone Pathway Analysis of Consistent DEGs and Functional Analysis of OsIAA3

Hormones, especially ABA and GAs, play a critical role in the regulation of seed germination. We analyzed the consistent DEGs of hormone pathways in the embryo and endosperm during seed germination. We found 5, 2, 2, 2, 25, 8, and 7 DEGs in the ABA, brassinosteroid (BR), cytokinin (CK), auxin (IAA), jasmonic acid (JA), salicylic acid (SA), and ethylene (ETH) pathways, respectively. This indicates that seed germination is regulated by multiple hormones, some of which may play major roles (i.e., ABA), and others potentially play auxiliary roles (i.e., ETH) (Figure 5A). Expression analysis showed that the expression patterns of hormone pathway genes were different in the embryo and endosperm during seed germination (Figure 5A). For example, the expression patterns of five genes in the ABA pathway were different in the embryo and endosperm. *OsPYL7*, *OsPYL8*, and *LOC9272602* were upregulated in the embryo and downregulated in the endosperm. *OsPYL4* was upregulated in the endosperm but downregulated in the embryo; *OsPYL9* was first downregulated and then upregulated in the endosperm, and downregulated continuously in the embryo (Figure 5A). Up to now, there has been a large amount of genetic evidence proving that several ABA pathway-related genes participate in the regulation of seed germination, but only a few IAA pathway-related genes have been reported to be involved in the regulation of this process. Therefore, we selected a gene from the consistently differentially expressed genes for functional verification, *OsIAA3* (*LOC4352722*), which was upregulated at 1 h and downregulated at 3 h until 9 h in the endosperm and downregulated at 1 h and then upregulated from 3 h to 24 h in the embryo during seed germination. Similarly, *iaa3* mutants were generated by CRISPR/Cas9, which inserted an “A” after the 148th nucleotide (Figure 5B), causing a frameshift and prematurely terminated, with 175 aa, while the wild-type protein contains 197 aa in Zhonghua 11 (ZH11) (Figure 5C). We found the germination process was slowed in the *iaa3* mutant (Figure 5D,E). A total of 30% of *iaa3* seeds germinated at 3 days after treatment with water, in contrast to 64.6% of WT seeds. 

### 2.5. Expression Analysis of Sugar Signaling and UPR Pathway-Related DEGs and Functional Analysis of OsBiP2

The metabolism of starch and sugar has been reported to be important changes in seed germination. KEGG analysis showed that the sugar-metabolism pathway was enriched in both the embryo and endosperm during seed germination. We analyzed the expression of starch degradation/synthesis and sugar-metabolism pathway genes, and the results showed that 3, 2, and 21 DEGs were identified in the starch degradation/synthesis and sugar metabolism pathway, respectively. Most of these genes have different expression patterns in the embryo and endosperm (i.e., *OsqGC-6*) (Figure 6A), similar to hormone signaling pathways. The UPR pathway is involved in the regulation of seed endosperm development, but its role in the regulation of rice-seed germination is not clear. We analyzed the expression of 12 UPR-pathway genes and found that these genes were all induced by seed germination, and some genes had similar expression patterns in the embryo and endosperm (i.e., *OsbZIP17*) while others did not (i.e., *OsBiP5*) (Figure 6B). In order to further clarify the function of the UPR pathway in rice-seed germination, we obtained a *bip2* mutant by using the CRISPR/Cas9 system in the japonica Nipponbare (NIP) background. An “A” was inserted after the 53rd nucleotide in the *bip2* mutant, causing a frameshift and prematurely terminated protein with 17 aa while the wild-type protein contained 670 aa in NIP (Figure 6C,D). We found that the germination speed was decreased in the *bip2* mutant, with 59% of *bip2* mutant seeds germinating at 3 days after treatment, in contrast to 95% of WT seeds (Figure 6E,F). 

### 2.6. Validation of DEGs by qRT-PCR

The expression level of the DEGs was verified by qRT-PCR. Eight DEGs were chosen for validation by qRT-PCR, including two genes in the ETH pathway (*ETR2*: *LOC4335058* and *EIL2*: *LOC4344331*), one gene in the IAA pathway (*AUX1*: *LOC4324749*), one gene in the BR pathway (*BZR1*: *LOC4343719*), three genes in the starch-degradation pathway (*Ramy1A*: *LOC4330832*, *Amy1c*: *LOC4330830* and *Amy3b*: *LOC9271949*), and one gene in the sugar-metabolism pathway (*RSUS2*: *LOC4340386*) (Appendix A). The expression patterns of these DEGs were basically consistent with the RNA-seq results, further confirming the reproducibility of the data (Appendix A).

## 3. Discussion

Omics analyses such as transcriptome, proteome, and metabolome analyses, have been used to reveal the process of seed germination in several crop species [8,41]. In rice, various omics methods have also been used to study seed germination. For example, Yang et al. (2020) conducted transcriptome and metabolome analyses of two types of rice to investigate seed germination and young seedling growth and revealed the involvement of cell-wall metabolism, lipid metabolism, and the ROS pathway in these processes [42]. Zhao et al. (2020) used transcriptome analysis of 8 h imbibed seeds and found that the stress response pathway is involved in the initial imbibition stage of rice-seed germination [7]. In summary, previous studies showed that a variety of biological processes are involved in rice-seed germination, including cell-wall metabolism, nucleotide degradation, amino acid biosynthesis, the stress-response pathway, and more. However, these results were obtained using shelled rice seeds, and the differences between the embryo and endosperm during germination were not distinguished. In our study, we analyzed the gene changes in the embryo and endosperm of unshelled rice seeds during germination using RNA-Seq. We found that the plant-hormone signal-transduction pathway mainly occurs in the embryo, while amino acid metabolism mainly occurs in the endosperm (Figure 2). This indicates that plant-hormone signal transduction has a specific role in embryos, while essential amino acid metabolism is important for the endosperm. We also found that during the early stage (0–3 h) of seed germination, the number of DEGs in the embryo is higher compared to that in the endosperm. However, the number of DEGs in the endosperm is higher compared to that in the embryo, during the late stage (12–24 h) of seed germination (Figure 3). Based on these results, we speculate that the embryo might respond earlier during seed germination than the endosperm, but the endosperm showed more genes involved in molecular regulation in seed germination at the intermediate and late stages compared to the embryo.

Seed germination is a complex process that is controlled by multiple genes, and so far, only a limited number of genes related to rice-seed germination have been characterized. At present, rice-seed germination-related genes have been identified mainly through map-based cloning and genome-wide association analysis (GWAS) [43]. For example, *qLTG3-1*, which controls low-temperature germinability, was cloned by map-based cloning using backcross inbred lines (BILs) derived from a cross between Italica Livorno and Hayamasari [44]. *OsSAP16* (stress-associated protein 16), which codes a zinc-finger domain protein associated with low-temperature germinability, was identified by GWAS with 187 rice natural accessions [45]. In addition, some QTLs have also been identified for seed germination. For instance, Dimaano et al. (2020) identified 43 QTLs governing early germination and seedling vigor using 167 BC1F5 selective introgression lines developed from a backcross population involving weed tolerant rice-1 as the recipient parent and Y-134 as the donor parent [46]. Furthermore, omics analysis is also a method to obtain the genes related to seed germination. For example, Zhao et al. (2021) found that *OsSAUR33* was highly expressed in both embryos and endosperm during seed germination by analyzing the *OsSAUR* gene family, and functional analysis showed that disruption of *OsSAUR33* resulted in reduced germination rates and low seed uniformity in early germination [47]. In this study, we identified two genes, *OsIAA3* and *OsBiP2*, through transcriptome analysis during seed germination. Functional analysis revealed that the knockout of *OsIAA3* and *OsBiP2* would lead to slower seed germination (Figure 5 and Figure 6). Therefore, transcriptome analysis is also an important method to obtain genes related to seed germination.

Since crosstalk between embryo and endosperm is important for seed germination, this study aimed to identify genes involved in seed germination regulation by analyzing differential genes in embryos and endosperm during this process. Multiple genes, including different TF families and hormone-pathway genes that may play a role in seed-germination regulation were screened (Figure 4 and Figure 5). The expression of some genes was verified by qRT-PCR (Appendix A). BZR1, a key TF in BR signaling, was found to be upregulated in the embryo and the endosperm during seed germination (Appendix A). BZR1 is known to play a crucial role in seed development and germination [48,49]. For example, ZmBZR1-5 positively regulates grain size and starch breakdown in maize [48]. Additionally, we verified the expression of two ETH pathway genes, ETR2 and EIL2, which were upregulated and then downregulated during seed germination (Appendix A). ETR2 is involved in regulating starch accumulation and seed germination [50,51,52], and the expression pattern of EIL2 and ETR2 is similar. However, more genetic evidence is required to show whether EIL2 is involved in regulating seed germination.

Auxin is a well-known phytohormone involved in various physiological processes, including seed dormancy and germination. Studies have shown that auxin inhibits seed germination, while mutations in auxin biosynthesis or receptor genes can release seed dormancy [9,53]. ARFs (auxin response factors) are also involved in seed dormancy and germination. In *Arabidopsis*, the negative regulation of ARF10 by miR160 is important for seed germination and post germination development; the mARF10 mutant seeds were hypersensitive to ABA during germination [54]. Auxin maintains *ABI3* expression by recruiting ARF10/16, thereby enhancing ABA-mediated seed dormancy during seed imbibition [9]. In rice, *OsIAA3* is a member of the *IAA* gene family, and its expression is rapidly increased in response to auxin. The *mOsIAA3-GR* transgenic rice is insensitive to auxin and exhibits abnormal leaf formation [55]. In this study, *OsIAA3* was found to be differentially expressed during seed germination and its expression pattern was different in embryos and endosperm. The seed-germination rate of the *OsIAA3* knockout mutant was slower compared with the wild type (Figure 5), indicating that *OsIAA3* plays an important role in regulating germination speed in rice. This study highlights the importance of transcriptome analysis in identifying seed-germination-related genes, such as *OsIAA3*.

The UPR pathway is evolutionarily conserved and plays essential roles in plant development and response to stress [56,57]. In *Arabidopsis*, the UPR pathway is involved in regulating seed development and response to ABA. *NAC103* is involved in the UPR pathway and is upregulated by ABA treatment. Overexpression of *NAC103* increases sensitivity to ABA during seed germination under exogenous ABA-treatment conditions [58]; *bZIP17* was also induced by ABA, and GFP-MYC-MbZIP17 was processed and translocated from the ER to the nucleus in response to ABA treatment, and expression of bZIP17△C rescued the ABA-sensitive phenotype of *s2p* mutants [59]. The above research indicated that the UPR pathway may regulate seed germination through the ABA pathway. The UPR pathway has also been shown to participate in protein synthesis and accumulation in the endosperm during seed development. For example, when seed-storage proteins are synthesized, folded, and assembled in the ER, *BiPs* are upregulated and *OsbZIP50* is spliced [60,61]. Several UPR-related genes have been identified in rice, including *OsBiP1*, *OsBiP2, OsBiP3*, *OsrAAT*, *OsbZIP50*, and *OsbZIP60*, which are involved in the occurrence of grain chalkiness [38]. In this study, some UPR-related genes, such as *OsBiP1* and *OsBiP2*, showed differential expressions during seed germination. Some genes showed similar (i.e., *OsBiP1*) or different (i.e., *OsBiP5*) expression patterns in embryos and endosperm. Interestingly, the seed germination rate of the *OsBiP2* mutant was slower than that of the wild type (Figure 6). All these results demonstrated that UPR is directly involved in the regulation of rice seed-germination.

In summary, our results provided new information on the understanding of gene responses in the embryo and endosperm during the seed germination of rice and provide insight into the effects of UPR on this germination process.

## 4. Materials and Methods

### 4.1. Plant Materials and Germination Experiments

The *Japonica* rice (*Oryza sativa* L.) cultivar NIP was used for seed-germination assay according to Zhao et al. (2020) [7], with modifications. For RNA sequencing, fifty unshelled seeds per replicate were imbibed in Petri dishes (d = 9 cm) with 10 mL double-distilled water in a growth chamber at 28 ± 1 °C for 24 h. We collected 5–8 seeds at 0, 1, 3, 6, 9, 12, and 24 h after treatment with double-distilled water, and the embryo and endosperm of each seed were separated. 

The *iaa3* and *bip2* mutants were generated using the CRISPR/Cas9 system in the ZH11 and japonica NIP background, respectively. Primers used for vector construction and mutant identification were listed in Appendix A.

For the germination experiments, the seeds were imbibed for 8 days under the same conditions as above, and the number of germinated seeds was recorded every day. Three biological replications were performed.

### 4.2. RNA Sequencing 

Total RNA was extracted from each sample using Trizol reagent (Invitrogen, Waltham, MA, USA) according to the manufacturer’s instructions, and the quality was checked by both the Qubit^®^ RNA Assay Kit in a Qubit^®^ 2.0 Fluorometer (Life Technologies, Carlsbad, CA, USA) and the Agilent^®^ 2100 bioanalyzer (Agilent Technologies, Santa Clara, CA, USA). Library construction was carried out according to the Illumina Stranded mRNA Prep kit (Illumina, San Diego, CA, USA) manufacturer’s instructions. The RNA samples were sequenced using the Illumina HiSeq 4000 platform by BGI Company, Shenzhen, China. An average of 6.7 gigabases of raw data for each sample were generated. Trimmomatic and Cutadapt were used to trim the low-quality reads and cut adapters to obtain clean reads for further analysis, respectively [62,63]. Clean reads were mapped to the reference genome of *Oryza sativar* GCF_001433935.1_IRGSP-1.0 using HISAT2 v2.2.1 [64]. Stringtie was used in the quantification of each sample [65], and edgeR was used to identify the differentially expressed genes [66]. Three biological replications were performed.

### 4.3. Differentially Expressed Gene Analysis 

The DEGs were identified with the criteria of *p*-value < 0.005 and |log_2_FC| > 1. The DEGs only differentially expressed at 1 h or 3 h after treatment were classified as the early-response genes; the genes only differentially expressed at 6 h or 9 h after treatment were regarded as intermediate-response genes; those only differentially expressed at 12 h or 24 h were late-response genes; and the genes differentially expressed at all stages were named as consistent DEGs. KEGG analysis was performed using the R package ‘clusterProfile’ with *p*-value < 0.05 and q-value < 0.05 [67]. The heatmap was generated using the ‘pheatmap’ function in R (https://www.rdocumentation.org/packages/pheatmap/, accessed on 5 March 2023). The TFs were analyzed based on the annotation of PlantTFDB (http://planttfdb.gao-lab.org, accessed on 15 February 2023). The Sankey diagram was generated using the function ‘sankeyNetwork’ in R (https://www.rdocumentation.org/packages/networkD3, accessed on 10 February 2023).

### 4.4. Quantitative Real-Time PCR Analysis

Total RNA was extracted from each sample using a Trizol reagent. The first-strand cDNA was synthesized, and qRT-PCR was carried out according to Xu et al. (2020) [68]. All selected genes were used for SYBR green real-time RT-PCR, and the primers are listed in Appendix A. The rice *UBQ10* gene was used as the internal control for normalization. The PCR reaction procedure was as follows: 95 °C for 5 min, followed by 40 cycles of 95 °C for 15 s and 60 °C for 30 s. The relative value of gene expression was derived from 2^−∆∆CT^ [69]. Three independent biological replications were performed for each sample.

## Figures and Tables

**Figure 1 ijms-24-08710-f001:**
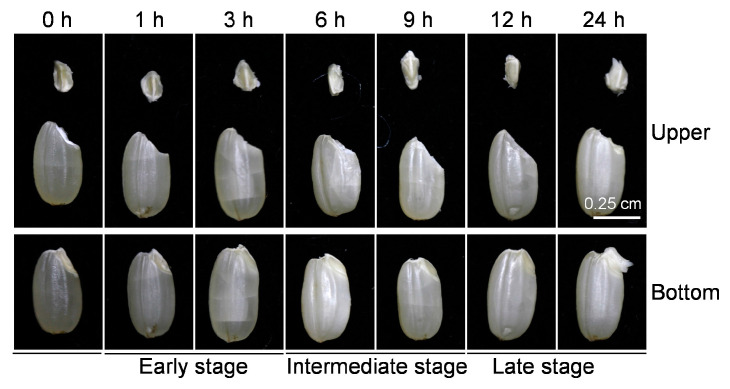
Dynamic changes during unshelled-rice-seed germination. The upper panel and bottom panel represent the separated embryo and endosperm and the whole seed after water treatment, respectively. Bar = 0.25 cm.

**Figure 2 ijms-24-08710-f002:**
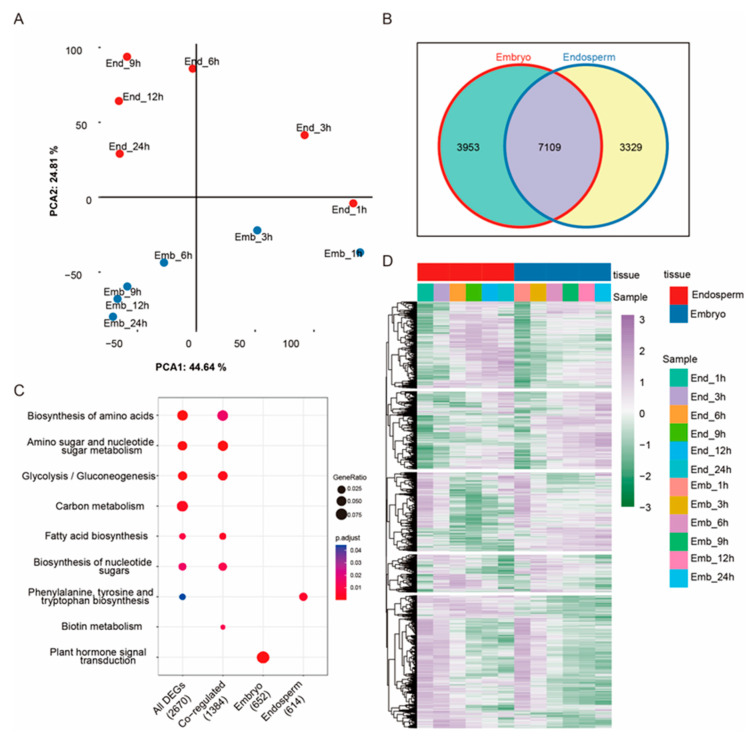
Transcriptomic profiling of germinating rice seed. The PCA analysis (**A**) and Venn diagram (**B**) of the embryo (navy point) and endosperm (red point) are presented. KEGG pathway enrichment with a *p*-value < 0.05 shows differences between embryo and endosperm (**C**). Heatmap representing expression patterns of DEGs in embryo and endosperm during seed germination (**D**). The left panel under the red bar represents DEGs in the endosperm (1 h, 3 h, 6 h, 9 h, 12 h, and 24 h), and the right panel under the navy bar represents DEGs in the embryo during seed germination (1 h, 3 h, 6 h, 9 h, 12 h, and 24 h).

**Figure 3 ijms-24-08710-f003:**
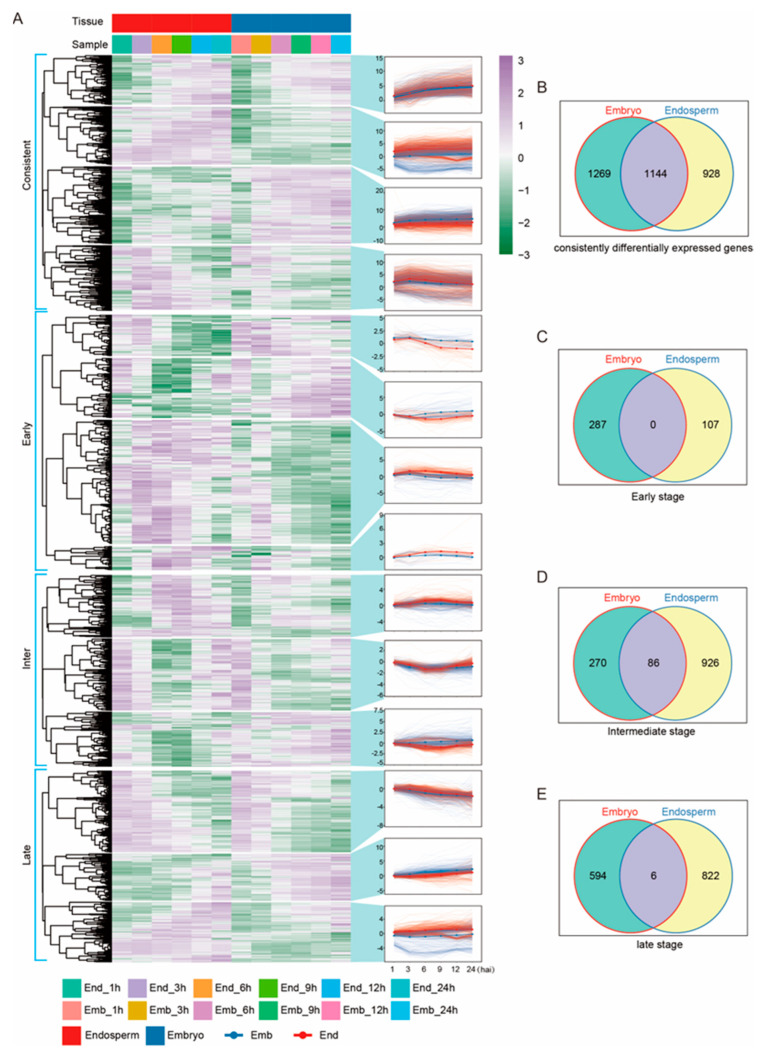
The temporal scale of gene expression models during rice-seed germination. Heatmap and hierarchical clustering of consistent-, early-, intermediate-, and late-response genes are shown in (**A**). The left panel under the red bar represents DEGs in the endosperm (1 h, 3 h, 6 h, 9 h, 12 h, and 24 h), and the right panel under the navy bar represents DEGs in the embryo during germination (1 h, 3 h, 6 h, 9 h, 12 h, and 24 h). Average gene expression trends of seed-germination response clusters of endosperm (red line) and embryo (navy line) are presented beside the heatmap. The number of DEGs in each cluster is shown in (**B**–**E**).

**Figure 4 ijms-24-08710-f004:**
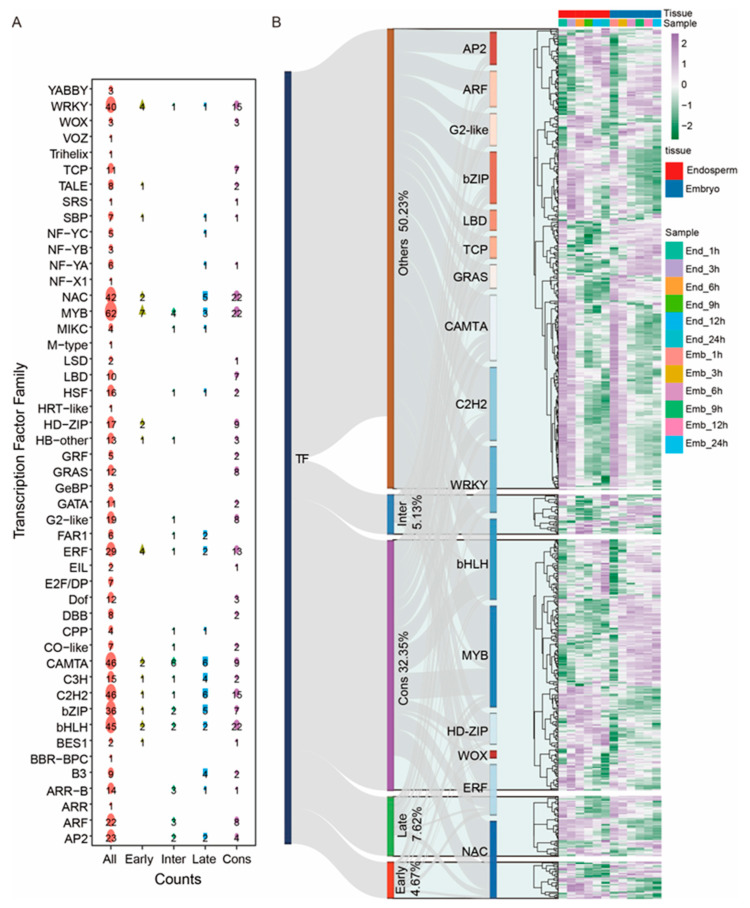
The identification of differentially expressed TFs during rice-seed germination. The number of TFs from each TF family is counted (**A**). The Sankey diagram represents a different percentage of TFs in each responsive stage and the related gene-expression patterns are on the right side (**B**). The most enriched top 16 TF gene families are shown in (**B**).

**Figure 5 ijms-24-08710-f005:**
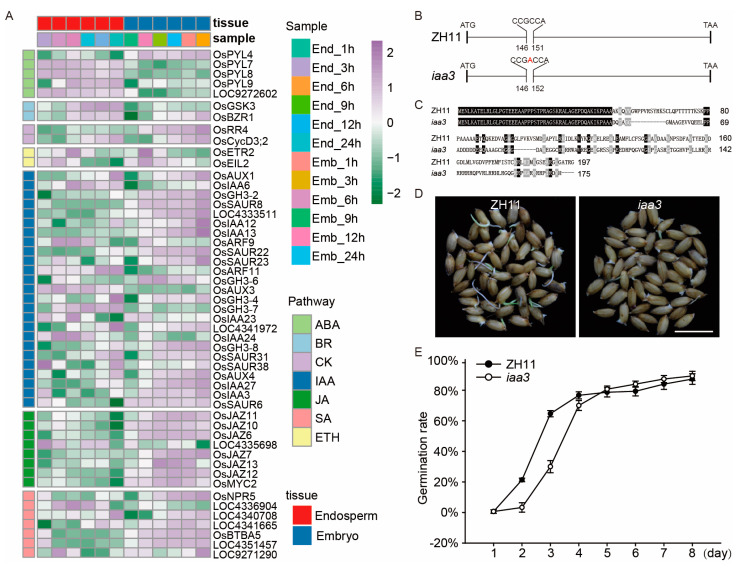
The expression patterns of the plant-hormone pathway genes during rice-seed germination and functional analysis of *OsIAA3*. The heat map of plant-hormone pathway genes (**A**), the expression value of each gene was based on the Z-score normalization value. The gene (**B**) and protein (**C**) structure of *OsIAA3* in wild-type (ZH11) and *iaa3* mutants. The red letter indicates the inserted nucleotide. The black and grey area represent amino acid conserved levels of 100% and larger than 50%, respectively. (**D**) Seed germination of the ZH11 and *iaa3* mutant under H_2_O treatment for 3 days. Bars = 1.0 cm. (**E**) Comparison of the germination rate between the ZH11 and *iaa3* mutant under H_2_O treatment. Error bars represent the SD from three biological replicates.

**Figure 6 ijms-24-08710-f006:**
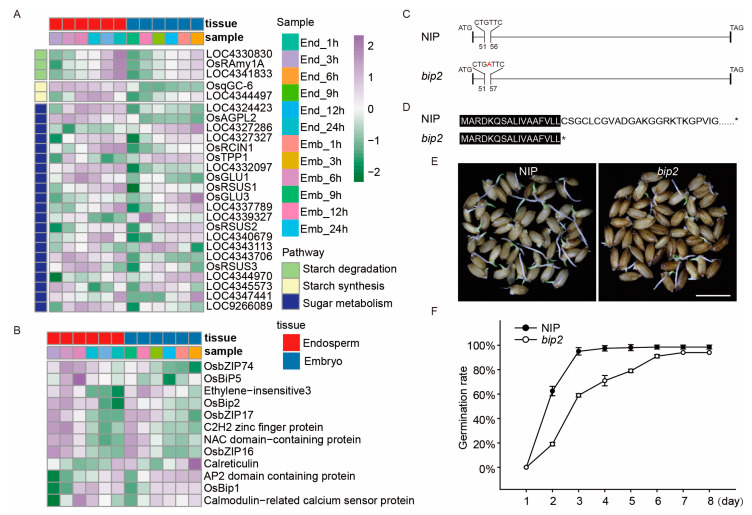
The expression patterns of starch and sugar metabolism and ER stress-related genes during rice seed germination and functional analysis of *OsBiP2*. The heat map of starch- and sugar-metabolism genes (**A**) and ER stress-related genes (**B**). The expression value of each gene was based on the Z-score normalization value. Gene (**C**) and protein (**D**) structure of *OsBiP2* in wild type (NIP) and *bip2* mutant. The red letter indicates the inserted nucleotide, The black area represents amino acid conserved levels of 100%. “*” indicates termination of translation. (**E**) Seed germination of the NIP and *bip2* mutant under H_2_O treatment for 3 days. Bars = 1.0 cm. (**F**) Comparison of the germination rate between the NIP and *bip2* mutant under H_2_O treatment. Error bars represent SD from three biological replicates.

## Data Availability

The raw sequence data reported in this paper have been deposited in the Genome Sequence Archive [70] in National Genomics Data Center [71], China National Center for Bioinformation/Beijing Institute of Genomics, Chinese Academy of Sciences (GSA: CRA010492) that are publicly accessible at https://ngdc.cncb.ac.cn/gsa (accessed on 20 March 2023).

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
