# Peer review of "Transcriptome Analysis of Rice Embryo and Endosperm during Seed Germination"

_ijms, 2023, doi:10.3390/ijms24108710_

Round 1

Reviewer 1 Report

see PDF

Author Response

Reviewer 1

  1. The draft by Zhang et al. (2023) titled Transcriptome analysis of rice embryo and endosperm during seed germination”, was reviewed at scientific and linguistic levels. The relationship between the embryo and the endosperm is an aspect studied during the germination process. Several publications have emerged in recent years about it. Likewise, the role of ABA, GAs and ethylene in this process was addressed by several authors. Together, to write an introduction on the subject of this interesting paper it is not necessary to go back 20 years. Therefore, I suggest that authors describe notable aspects of germination and do not resort to outdated events [e.g. water uptake by a mature dry seed is triphasic (references)]. The introduction should be halved. I miss much more current references in the introduction itself. Authors such as Nonogaki, H., Leubner-Metzger, G., Penfield, S., Nambara, E., Matilla, A., Seo, M., Marion-Poll, A., Liu, X., Soppe, W.J., Bentsink, L., among others, are some examples.

Reply 1: Thank you, we rewrote the introduction of the manuscript and referred to some current references.

Mayor

  1. Lines 61-85. A very important percentage of this text has nothing to do with the objectives of this work.

Reply 2: We rewrote this part in lines 67-108.

  1. Line 66…. [12,14, and doi: 10.3389/fpls.2022.1000803, among others….]

Reply 3: We have added the references in line 72.

  1. For what devote so much space to x-amylase?

Reply 4: We rewrote this part in lines 67-108.

  1. Lines 86-99. Wording, english and bibliography have not adequately revised. If we add to this a lack of clear concepts about the germination process, a rather wordy second part of the Introduction appears.

Reply 5: We have asked MDPI English editing service to proofread the manuscript, and rewrote this part.

  1. Line 103. … (for example BiPs et. al)….. This error is indicative of the lack of revision of the text.

Reply 6: We have revised in lines 127-129.

  1. Lines 100-134. Again, it makes no sense that the authors have made such a long and tedious introduction. This draft is not an update...

Reply 7: We have revised this part in lines 123-160.

  1. Lines 125-127. ….. We found that the embryo and endosperm play different functions, and the expression patterns of some genes are different in the embryo and endosperm during seed germination. Oh my God!!!!....

Reply 8: We have revised this paragraph in lines 150-151.

  1. The last part of Introduction suffers from the same criticism as the rest……

Reply 9: We have revised this part in lines 123-160.

Minor

  1. Line 36. ….. plant development and crop yield [1].

Reply 10: We have revised in line 36.

  1. Line 37. …… and energy metabolisms.

Reply 11: We have revised in line 38.

  1. Line 41. …… more and more attention (rewrite; verb is missing).

Reply 12: We have revised this paragraph in lines 42-43.

  1. Line 45. ….. many procedures have been used….

Reply 13: We have revised in line 46.

  1. Lines 46-49. Water uptake by a mature dry seed is triphasic: (i) rapid initial uptake (Phase I); plateau phase (Phase II), and (iii) further increase in water uptake after germination is completed (Phase III).

Reply 14: We have revised in lines 48-50.

  1. Line 50. ….. antioxidant-related… (something is missing here).

Reply 15: We have revised this paragraph in lines 52-54.

  1. Line 51. …… indicating that in this phase it takes place preferably the activation…..

Reply 16: We have revised in lines 54-56.

  1. Line 52. ….. at the beginning of Phase II …..

Reply 17: We have revised in line 57.

  1. Line 53. ….. ; in the middle stage of Phase II….

Reply 18: We have revised in line 58.

  1. Line 55. …… involving energy metabolism, cell growth, cell defense and storage proteins are activated [8]. (something is missing here)

Reply 19: We have revised this paragraph in line 58-60.

  1. Line 55. …… and at the end of Phase II….

Reply 20: We have revised in lines 60-61.

  1. Line 58. …… scarce research….

Reply 21: We have revised in line 64.

  1. Lines 61-63. Beware, those factors are not genetic!!!!!

Reply 22: Thank you, we have revised in line 67.

  1. Line 63. …… phytohormones that antagonistically regulate seed germination [10, 11, other more recent]. GA and ABA play antagonistic roles in regulating seed germination, with ABA suppress- 64 ing and GA enhancing seed germination

Reply 23: We have revised in lines 69-72

  1. Line 65. As ABI3, …. As?

Reply 24: We have revised in lines 83-90.

Results

  1. Lines 137-138. To determine…. This phrase makes no sense.

Reply 25: We have revised in lines 149-150.

  1. Figs. 2A, 2B, 2C, 2D do not exist.

Reply 26: We reversed the order of figures 1 and figure 2 in the previous manuscript, and we have revised the order and legends of Figure 1 and Figure 2.

  1. Fig. 2 (legend). Does not correspond to Fig. 2.

Reply 27: We reversed the order of figures 1 and figure 2 in the previous manuscript, and we have revised the order and legends of Figure 1 and Figure 2.

Reviewer 2 Report

Review of the article:

 Transcriptome analysis of rice embryo and endosperm during seed germination

Summary

14,391 DEGs of the Japonica rice (Oryza sativa L.) cultivar Nipponbare (NIP) were identified between dry seeds of rice (0-hour time point) and after unshelled rice seed imbibition. About half of them are expressed both in the embryo and endosperm while 3953 are embryo specific and 3329, endosperm specific. The analysis is concentrated on the expression of genes encoding transcription factors and on the UPR pathway genes were induced by seed germination.

General comment

The work is technically competent. The main originality is that the analysis of endosperm and embryo is done separately. It does not provide very original results. English language needs a review and correction.

Comments by sections

Introduction

The English language needs polishing.

Correct:

Such as in Arabidopsis, knock-outs of AtbZIP28 and At-bZIP60 confer high sensitivities to heat stress at reproductive stages in terms of silique length and seed setting[29]. In rice, OsbZIP74 and OsNTL3 are constitutively expressed and up-regulated by heat and ER stresses, and co-regulate the response of rice to heat stress[30].

To:

In Arabidopsis, knock-outs of AtbZIP28 and At-bZIP60 confer high sensitivities to heat stress at reproductive stages in terms of silique length and seed setting[29]. In rice, OsbZIP74 and OsNTL3 are constitutively expressed and up-regulated by heat and ER stresses, and co-regulate the response of rice to heat stress [30].

BiP is introduced suddenly and without an adequate presentation. Please explain what is BiP.

Results

The English language needs polishing.

2.1. Transcriptome Sequencing and Differentially Expressed Gene Analysis

Please correct:

To determine the difference of gene responses in embryo and endosperm during seed germination in rice. The rice unshelled seeds were imbibed with double distilled water, and sampled at 0, 1, 3, 6, 9, 12 and 24 hours after imbibition (Figure 1).

To:

To determine the difference of gene responses in embryo and endosperm during seed germination in rice the rice unshelled seeds were imbibed with double distilled water, and sampled at 0, 1, 3, 6, 9, 12 and 24 hours after imbibition (Figure 1).

Mention to Figure 1 is not appropriate here.

Based on seed imbibition, we divide the seed germination process into three stages: the first stage is associated with rapid water uptake (early stage, 0-3 h), the second stage is associated with embryo expansion (intermediate stage 6-9 h), and the third stage is the beginning of germination when the radicle begins to grow (late stage 12-24 h) (Figure 1).

(Mention to Figure 1 is not appropriate here neither. Nothing in this paragraph is illustrated by Figure 1).

Legend to Figure 1: There is a lot of information missing. You have to explain the contents of Figure 1 in the legend to the figure and in the text.

In the present version there is a confusion in the figure legends to Figure 1 and Figure 2. This has to be revised. All the text must contain information pertinent to the figures and vice-versa. The figures must be ordered each immediately below the corresponding text.

2.4. Hormones Pathway Analysis of consistently DEGs and Functional Analysis of OsIAA3

Change:

Expression analysis showed that the expression patterns of hormone pathway genes were inconsistent in the embryo and endosperm during seed germination (Figure 5A).

To:

Expression analysis showed that the expression patterns of hormone pathway genes were different in the embryo and endosperm during seed germination (Figure 5A).

2.5. Expression Analysis of Sugar Signaling and UPR Pathway Related DEGs, and Functional 266

Analysis of OsBiP2

Most of these genes have inconsistent expression patterns in the embryo and endosperm (i.e. OsqGC-6) (Figure 6A), similar to hormone signaling pathways.

Correct to:

Most of these genes have different expression patterns in the embryo and endosperm (i.e. OsqGC-6) (Figure 6A), similar to hormone signaling pathways.

Correct:

In order to further clarify the function of the UPR pathway in rice seed germination. We obtained bip2 mutant by using the CRISPR/Cas9 system in the japonica Nipponbare (NIP) background.

To:

In order to further clarify the function of the UPR pathway in rice seed germination, we obtained bip2 mutant by using the CRISPR/Cas9 system in the japonica Nipponbare (NIP) background.

Discussion

Correct:

Seed germination is a quantitative trait controlled by multiple genes.

Seed germination is a complex process controlled by multiple genes.

Revise this paragraph:

For example, qLTG3-1, For germination rate under various con-ditions has been cloned in rice by map-based cloning using backcross inbred lines (BILs) derived from a cross between Italica Livorno and Hayamasari[42].

Author Response

Reviewer 2

General comment

  1. The work is technically competent. The main originality is that the analysis of endosperm and embryo is done separately. It does not provide very original results. English language needs a review and correction.

Reply 1: Thank you. The raw sequence data reported in this paper have been deposited in the Genome Sequence Archive in National Genomics Data Center, China National Center for Bioinformation / Beijing Institute of Genomics, Chinese Academy of Sciences (GSA: CRA010492) that are publicly accessible at https://ngdc.cncb.ac.cn/gsa, and we have asked MDPI English editing service to proofread the manuscript.

  1. Correct:

Such as in Arabidopsis, knock-outs of AtbZIP28 and At-bZIP60 confer high sensitivities to heat stress at reproductive stages in terms of silique length and seed setting[29]. In rice, OsbZIP74 and OsNTL3 are constitutively expressed and up-regulated by heat and ER stresses, and co-regulate the response of rice to heat stress[30].

To:

In Arabidopsis, knock-outs of AtbZIP28 and At-bZIP60 confer high sensitivities to heat stress at reproductive stages in terms of silique length and seed setting[29]. In rice, OsbZIP74 and OsNTL3 are constitutively expressed and up-regulated by heat and ER stresses, and co-regulate the response of rice to heat stress [30].

Reply 2: We have revised in the introduction.

  1. BiP is introduced suddenly and without an adequate presentation. Please explain what is BiP.

 Reply 3: We have introduced the BiP when it first appeared.

  1. 2.1. Transcriptome Sequencing and Differentially Expressed Gene Analysis

Please correct:

To determine the difference of gene responses in embryo and endosperm during seed germination in rice. The rice unshelled seeds were imbibed with double distilled water, and sampled at 0, 1, 3, 6, 9, 12 and 24 hours after imbibition (Figure 1).

To:

To determine the difference of gene responses in embryo and endosperm during seed germination in rice the rice unshelled seeds were imbibed with double distilled water, and sampled at 0, 1, 3, 6, 9, 12 and 24 hours after imbibition (Figure 1).

Reply 4: We have revised in 2.1.

  1. Mention to Figure 1 is not appropriate here.

 Reply 5: We have changed the mention to Figure1.

  1. Based on seed imbibition, we divide the seed germination process into three stages: the first stage is associated with rapid water uptake (early stage, 0-3 h), the second stage is associated with embryo expansion (intermediate stage 6-9 h), and the third stage is the beginning of germination when the radicle begins to grow (late stage 12-24 h) (Figure 1).

(Mention to Figure 1 is not appropriate here neither. Nothing in this paragraph is illustrated by Figure 1).

Reply 6: We reversed the order of figures 1 and figure 2 in the previous manuscript, and we have changed the mention to Figure1.

  1. Legend to Figure 1: There is a lot of information missing. You have to explain the contents of Figure 1 in the legend to the figure and in the text.

 Reply 7: We have revised the legend and description in manuscript of Figure 1.

  1. In the present version there is a confusion in the figure legends to Figure 1 and Figure 2. This has to be revised. All the text must contain information pertinent to the figures and vice-versa. The figures must be ordered each immediately below the corresponding text.

 Reply 8: We have revised the order and legends of Figure 1 and Figure 2.

  1. 2.4. Hormones Pathway Analysis of consistently DEGs and Functional Analysis of OsIAA3

Change:

Expression analysis showed that the expression patterns of hormone pathway genes were inconsistent in the embryo and endosperm during seed germination (Figure 5A).

To:

Expression analysis showed that the expression patterns of hormone pathway genes were different in the embryo and endosperm during seed germination (Figure 5A).

Reply 9: We have revised in 2.4.

  1. 10. 2.5. Expression Analysis of Sugar Signaling and UPR Pathway Related DEGs, and Functional Analysis of OsBiP2

Most of these genes have inconsistent expression patterns in the embryo and endosperm (i.e. OsqGC-6) (Figure 6A), similar to hormone signaling pathways.

 Correct to:

Most of these genes have different expression patterns in the embryo and endosperm (i.e. OsqGC-6) (Figure 6A), similar to hormone signaling pathways.

Reply 10: We have revised in 2.5.

  1. Correct:

In order to further clarify the function of the UPR pathway in rice seed germination. We obtained bip2 mutant by using the CRISPR/Cas9 system in the japonica Nipponbare (NIP) background.

 To:

In order to further clarify the function of the UPR pathway in rice seed germination, we obtained bip2 mutant by using the CRISPR/Cas9 system in the japonica Nipponbare (NIP) background.

 Reply 11: We have revised in lines 313-314.

  1. Discussion

Correct:

Seed germination is a quantitative trait controlled by multiple genes.

Seed germination is a complex process controlled by multiple genes.

Reply 12: We have revised in Discussion.

  1. Revise this paragraph:

For example, qLTG3-1, For germination rate under various con-ditions has been cloned in rice by map-based cloning using backcross inbred lines (BILs) derived from a cross between Italica Livorno and Hayamasari[42].

 Reply 13: We have revised this paragraph.

Reviewer 3 Report

General comments:

Overall, this manuscript provides a comprehensive analysis of the gene expression changes that occur during seed germination in rice. The authors identify many differentially expressed genes (DEGs) and categorize them into various stages and functional pathways. They also perform transcription factor analysis and identify several key genes that are involved in the unfolded protein response (UPR) pathway and affect seed germination rates.

The manuscript is well-written and clearly presented. However, I have some comments for the discussion of the data before going further.

1.     Firstly, the authors discuss the functional enrichment of DEGs, they do not provide any detailed discussion or interpretation of the results. It would be useful to have a more in-depth analysis and discussion of the functional pathways identified, and how they may be related to seed germination.

For example, the paper shows some known DEGs that are closely connected to each other for seed development, In Figure 4, the AP2 TF has been reported in seed development and in Figure S2 the author validates some genes like BZR1, the key TF in BR signaling, and Amy sugar-related(starch) genes. And it is very interesting that the BZR1 has been reported to regulate the AP2 to regulate the seed development (PMID: 33206180). This BZR1 in maize has another domain(amylase) related to sugar metabolism with recent structure papers (PMID: 35961473), which indicated that the amylase domain is a putative regulatory domain and/or metabolic sensors in plants.

All these connected genes should be very interesting to discuss to be consistent with the data presented and must be discussed with the meaning of the study here

2.     Line 17. after in this study, the method should be shortly described here.

3.     Line 296 the gene name should be included here like BZR1, known key TFs in BR signaling.

4.     The method must be improved, please indicated more details.

Author Response

 Review3

General comments:

Overall, this manuscript provides a comprehensive analysis of the gene expression changes that occur during seed germination in rice. The authors identify many differentially expressed genes (DEGs) and categorize them into various stages and functional pathways. They also perform transcription factor analysis and identify several key genes that are involved in the unfolded protein response (UPR) pathway and affect seed germination rates.

The manuscript is well-written and clearly presented. However, I have some comments for the discussion of the data before going further.

  1. Firstly, the authors discuss the functional enrichment of DEGs, they do not provide any detailed discussion or interpretation of the results. It would be useful to have a more in-depth analysis and discussion of the functional pathways identified, and how they may be related to seed germination.

For example, the paper shows some known DEGs that are closely connected to each other for seed development, In Figure 4, the AP2 TF has been reported in seed development and in Figure S2 the author validates some genes like BZR1, the key TF in BR signaling, and Amy sugar-related(starch) genes. And it is very interesting that the BZR1 has been reported to regulate the AP2 to regulate the seed development (PMID: 33206180). This BZR1 in maize has another domain(amylase) related to sugar metabolism with recent structure papers (PMID: 35961473), which indicated that the amylase domain is a putative regulatory domain and/or metabolic sensors in plants.

All these connected genes should be very interesting to discuss to be consistent with the data presented and must be discussed with the meaning of the study here

 Reply 1: Thank you, we have added the discussion section in lines 387-400.

  1. Line 17. after in this study, the method should be shortly described here.

 Reply 2: We have revised in abstract.

  1. Line 296 the gene name should be included here like BZR1, known key TFs in BR signaling.

 Reply 3: We have added the gene name in 2.6.

  1. The method must be improved, please indicated more details.

 Reply 4: We have revised the method in Materials and Methods.

Round 2

Author Response

Reviewer 1

REPORT 2

Zhang et al. Transcriptome analysis of rice embryo and endosperm during seed germination. Int. J. Mol. Sci. 2023

For the authors,

my Report 1 was very clear. That is, if authors did not carry out a strong review of the Introduction, Figs. and Ref. I was not willing to accept this paper. Now, I have no choice but to add results and discusión.

  1. Thus, at least I will find 44 errors in references (e.g. nº 9, Proceedings of the National Academy of Sciences of the United States of America; nº 19, The Plant journal: for cell and molecular biology; nº 30, Lepidium sativum (cress); nº 36, Journal missing, etc. etc.). Again, very careless text.

Reply 1: We have revised the references.

  1. Summary: line 18, …. a total of 14,391 DEGs were identified…. What is DEGs?? Authors should know that the first time an abbreviation is cited, the meaning must be indicated [i.e. differentially expressed genes (DEGs)].

Reply 2: Thank you, we have revised in line 18.

  1. Line 19, … 0-hour time point… (physiological error) What is? Perhaps dry seed??

Reply 3: Thank you, we have revised in line 19.

  1. Line 19, the concept “differentially” is correctly used??

Reply 4: Thank you, we have revised in line 19.

  1. Line 28, The unfolded protein response (UPR) pathway genes… The Summary is very important in any paper; and this is the second version....

Reply 5: We have revised the abstract.

  1. Introduction. Line 48, comprising (delete). Lines 69-77; numbers [10-16], where are they placed in text? Why do the authors exclusively emphasize the ABA? Why do the authors prioritize sugars and x-amylase over auxins, ethylene or GAs here?

Reply 6: The references [10-16] has been removed in the first revision. Here, we present that different hormones and sugar are involved in regulating seed germination because our results involve these factors, for example 2.4 and 2.5 in the results section.

  1. Sentence 111-112, is not correct at physiological level. Mechanical constraint by which organ?

Reply 7: We have revised in line 118.

  1. Line 116, α-amylase.

Reply 8: We have revised in line 123.

  1. Line 118, ref. [31] is very old.

Reply 9: Thank you, we have replaced the reference mentioned.

  1. Line 120, …. “cell expansion”…. from whom?

Reply 10: We have revised in line 127.

  1. Lines 124-125, … “secreted and membrane proteins need to enter the endoplasmic reticulum”…. (rewrite).

Reply 11: We have revised this paragraph in lines 132-135.

  1. Lines 142-149, must be rewritten to clarify.

Reply 12: We rewrote this part in lines 150-158.

  1. Line 153, KEGG (Kyoto Encyclopedia of Genes and Genomes) analysis…

Reply 13: We have revised in line 168.

  1. Lines 153-161, must be rewriten to clarify at the seed physiological level (e.g. not completely consistent during seed germination; for example; and others…).

Reply 14: We rewrote this part in lines 168-180.

  1. Results. Some defect and mistakes. "After imbibition" (lines 151, 165-166; 198, 202, 206, 208, 239, 242, 449-451, 478, etc. etc.) means that the imbibition process has finished. Which is not true in Fig. 1. It would be correct if the authors started from imbibid seeds and put them to germinate 1-24 h.

Reply 15: Thank you, we have made revisions to the use of "After imbibition" throughout the entire text.

  1. Lines 455-457: notice, here the seeds used to germinate are seeds imbibid for 8 d. Thus, Fig. 1 is very bad explained. Legend (Fig. 1): Early stage: 1-3 hai, intermediate stage: 6-9 hai, late stage: 12-24 h. This already is in the Fig. 1 (i.e. repetition).

Reply 16: Thank you, we have revised the legend of figure1.

  1. Lines 175-87: (i) 14391, 7109 and…, thousands are separated by comma (see summary); (ii) “differentially expressed” (lines 177-78), is not correct ; (iii) you should consider that the “enriched” concept is referred to genes.

Reply 17: We have revised the representation of numbers throughout the entire text, modified the use of " differentially expressed," and revised this paragraph in lines 198-207.

  1. Line 212: “… that signaling in the embryo may precede that in the endosperm in seed germination after imbibition” (this sentence is merely speculative).

Reply 18: Thank you, we have deleted this paragraph in results.

  1. Sentences 185-87, 218-221, 276-277, 289-291, 320-321…. These interpretations (hypothesis) should be placed in discussion, and not in results.

Reply 19: Thank you, we have moved this content to the discussion section.

  1. Line 343, For Example…. Lines 341-364, the authors use too much space to confirm us that this manuscript has different connotations than previous ones. Therefore, it should be shortened. ……. The discussion of a manuscript has enormous relevance. Not only as a clear demonstration that the authors demonstrate a knowledge of the subject under study, but also for its value in the scientific progress that the publication entails. This is not the case. The authors have not adequately focused the discussion and have not been able to highlight some interesting data from this draft. From my point of view, this work, with an important informatic component, does not have the merits to be published in this journal.

Reply 20: We have revised the discussion in lines 375-512.

Reviewer 2 Report

The article has been improved. Nevertheless, it still needs an English review.

Related to Figures 1 and 2, please make sure the whole content of Figure 1 is not repeated in figure 2.

Author Response

Reviewer 2

  1. The article has been improved. Nevertheless, it still needs an English review.

Reply 1: Thank you, we have asked MDPI English editing service and native English speaker to proofread the manuscript.

  1. Related to Figures 1 and 2, please make sure the whole content of Figure 1 is not repeated in figure 2.

Reply 2: Thank you, we have checked the content of Figures 1 and 2 and confirmed that they are not repeated.

Round 3

Author Response

Response to Reviewer 1 Comments

  1. I value very positively the effort of the authors to rewrite important parts of the text. Likewise, I also appreciate that they have accepted all my suggestions focused on improving the draft. Together, since the text has been substantially enriched, I give the go-ahead to be published in Int. J. Mol. Sci.

Reply 1: Thank you.

MINORS

  1. Initially (summary), transcription factor was abbreviated as TF. Therefore, TF should be used throughout the entire text.

Reply 2: We have revised the abbreviation of transcription factor throughout the entire text.

  1. 102 gibberellins (GAs)… IN THE ENTIRE TEXT

Reply 3: We have revised the “GAs” throughout the entire text.

  1. 115 …. the interaction between BES1, a central component TF of brassinosteroid signaling, ….

Reply 4: We have revised in lines 74-75.

  1. 775 The expresion pattern…..

Reply 5: We have revised in line 214.

  1. 783, 785, 1301 ethylene (ETH)….. … two ETH …

Reply 6: We have revised in lines 229 and 352.

  1. 799 Likewise, iaa3 mutants….

Reply 7: We have revised in lines 245.

  1. 28, 801, 949, 1319 wild type or wild-type

Reply 8: We changed wildtype to wild-type in the entire text.

  1. 751 Transcription factors TFs ….

Reply 9: We have revised in lines 208-209.

  1. 929 Bars = 0 cm. SEPARATION

Reply 10: We have revised in lines 255 and 283.

  1. 1557 …. The effects of UPR on this germination process.

Reply 11: We have revised in line 395.

  1. References:

1666 Cell

1668 Biochemical and Biophysical Research Communications

1680 Physiologia Plantarum

1742 Frontiers in Plant Science

1756 Annual Review of Plant Biology

1757 International Journal of Molecular Sciences

1760 The Plant Journal

1763, 1765 New Phytologist

Etc,etc . The main words of all Journals are capitalized.

Reply 12: Thank you, we have revised the references and capitalized the main words of the journals.
